# Evaluation of the Antimicrobial Capacity of Bacteria Isolated from Stingless Bee (*Scaptotrigona aff. postica)* Honey Cultivated in Açai (*Euterpe oleracea*) Monoculture

**DOI:** 10.3390/antibiotics12020223

**Published:** 2023-01-20

**Authors:** Iago Castro da Silva, Eveson Oscar Almeida Conceição, Daniel Santiago Pereira, Hervé Rogez, Nilton Akio Muto

**Affiliations:** 1Centre for Valorization of Amazonian Bioactive Compounds & Biological Science Institute, Federal University of Pará, Belém 6075-110, PA, Brazil; 2Embrapa Amazônia Oriental, Belém 66095-903, PA, Brazil

**Keywords:** plant-derived compounds, stingless bee honey, pathogenic strains, antimicrobial activity

## Abstract

Many antimicrobial compounds have been seeking to protect the human body against pathogenic microbial infections. In recent times, there has been considerable growth of pathogens resistant to existing drugs due to the inappropriate use of antibiotics. In the present study, bacteria isolated from the honey of stingless bees native to the Amazon called *Scaptotrigona aff. postica* and *Apis mellifera* were used to determine their potential antimicrobial properties and characterize the medium cultivated with isolated bacteria. The results showed inhibition of nine isolates. Among these isolates, SCA12, SCA13, and SCA15 showed inhibitory activity similar to that of vancomycin, which was used as a positive control. The SCA13 strain obtained the best results with antimicrobial extract against the tested pathogens; the species was identified as *Enterococcus faecalis,* and its lyophilized extract was characterized by temperature, pH, and trypsin, in which they showed antimicrobial activity. This work shows that bacteria isolated from the stingless bee honey, *Scaptotrigona aff. postica,* have the potential to produce antimicrobial substances.

## 1. Introduction

Bee honey has been studied for centuries, from classical antiquity to the present day. In this context, honey has been used for various purposes, ranging from food supplements to wound healing. Honey, along with other bee products, is associated with the image of a natural, healthy, and clean product [1]; its medicinal properties and antimicrobial activity are generally related to its physical and chemical characteristics [2]. Gonçalves et al. [3] mentioned that in addition to its nutritional properties, the use of honey in folk medicine is also due to its pharmacological properties. Among these properties, antimicrobial activity has attracted interest among researchers because of its potential applicability in clinical cases [4,5,6].

The honey of stingless bees, known as meliponines, is used in popular therapies, mainly in rural areas [7] and among indigenous peoples, who believe that different types of honey have specific healing properties and are used to cure a wide range of diseases [8]. Understanding the antibacterial potential of bee honey and its microorganisms can be an attractive and low-cost alternative for the treatment of clinical conditions, in addition to leveraging the production chain of the stingless bee product, *Scaptotrigona aff. postica* [9]. Bees of the Meliponini tribe comprise approximately 60 genera distributed throughout the tropical and subtropical regions of the world. They are important insects in the maintenance of ecosystems, acting as a source of food, in addition to dispersing seeds and pollen through pollination. This last function gives bees the name of pollinating insects, and among these is the species *Scaptotrigona depilis*, common in neotropical regions such as Argentina, Bolivia, Paraguay, and Brazil, with prevalence in the states of Mato Grosso do Sul, Minas Gerais, Paraná, Rio Grande do Sul, and São Paulo [10].

Among the pharmacological properties of honey, its antimicrobial activity, such as healing and antioxidant capacity, has aroused interest among researchers because of its potential applicability in clinical cases. Therefore, the use of honey as a therapeutic agent has shown promising results. However, therapeutic effects vary according to the constituents and compound varieties. Honey has been reported to have an inhibitory effect on approximately 60 species of bacteria, 3 including aerobic and anaerobic gram-positive and gram-negative bacteria, such as *Staphylococcus aureus* and *Pseudomonas aeruginosa*, which are considered opportunistic pathogens [11,12,13,14,15,16,17].

In general, honey produced by the meliponine species presents several differences in physicochemical terms compared to honey produced by *A. mellifera*. For instance, honey from stingless bees has higher acidity, which can be detected by its taste and sensorial properties [18]. Acidity is important for maintaining stability and reducing the risk of the development of pathogenic microorganisms. Physicochemical factors (such as pH and acidity) are considered important antimicrobial factors, providing greater stability to the product, since the optimal pH for the growth of several pathogens in animals ranging from 7.2 to 7.4. Thus, antimicrobial activity can be caused by several physicochemical factors, such as pH, humidity, and sugars present in its composition. In addition to physicochemical factors, the presence of symbiotic bacteria in honey detains mechanisms that produce peptides and other molecules with antimicrobial properties, such as organic acids [19].

It is now recognized that most honey has proven antibacterial activity, which is dependent on a variety of factors. These factors include hydrogen peroxide generation, osmosis, acidity, limited availability of water molecules, and the presence of antimicrobial molecules, such as flavonoids and phenolic acids. In addition to their proven antibacterial activity, flavonoids are powerful antioxidants with an incredible ability to scavenge free radicals [20]. Regarding microorganisms with antimicrobial properties found in honey, several bioactive molecules such as potential antibiotics are the results of secondary metabolites of bacterial fermentation processes, with other compounds that may present antioxidant activity [19,20].

Honey from açai (*Euterpe oleracea*) produced in Pará has different qualities from other wild floral nectar honey because the açai bioactive compounds are not only found in the fruit, but also in different vegetal organs of the plant, such as the floral nectar. Well-known bioactive compounds present in açai, such as anthocyanins, flavonoids, non-anthocyanin flavonoids, amino acids, fatty acids, and minerals, have important anti-inflammatory and antioxidative properties. In addition, açai fruits have high levels of total mesophilic bacteria with high diversity, some of them being probiotic candidates and antagonistic potential against pathogens [21].

Given the importance of studies cited on physicochemical factors and microbial composition of honey from stingless bees, the objective of the present study was to detect in vitro antimicrobial activity from bacteria isolated from the honey of the native Amazonian stingless bee *Scaptotrigona aff. postica* against pathogenic bacteria. In addition, we characterized the compounds produced by these bacteria in relation to oxygen concentration, temperature, pH, and enzymatic activity.

## 2. Results

### 2.1. Sample Isolation

Twenty-eight strains were obtained from 9 honey samples (5 of *A. mellifera* and 4 of *S. aff. postica*), 22 strains of which were isolated from *S. aff. postica* honey and 6 strains of which were isolated from *A. mellifera* honey. Microbial growth occurred on plates that were anaerobically inoculated with *A. mellifera* honey in MRS agar culture medium. The first honey concentration of 100% was pure, the second was a concentration diluted in 50% saline solution, and the third corresponded to a concentration diluted in 25% of honey in saline solution to evaluate whether the concentration of honey is a parameter that influences the number of colonies obtained. Preliminary results confirmed that there was growth only in the anaerobic plates in the spread plate (50% and 25%) and pour plate (100% and 50%) techniques, as shown in Table 1.

In the procedure performed with honey from *S. aff. postica*, many plates showed microbial growth, with high bacterial content which grew in 10^−^¹, 10^−^², and 10^−3^ dilutions of samples.

Figure 1 shows the antimicrobial test performed on samples of *S. aff. postica* honey at three concentrations (25%, 50%, and 100%) against *E. coli*. The results showed that the antimicrobial capacity of honey from *S. aff. postica* was higher at the highest concentration used. As can be seen in Figure 1, the inhibition halos are higher according to the honey concentration used. Based on these results, it was observed that the smallest halos were observed as the samples were diluted. The lower the concentration, the lower the antimicrobial activity of the honey, a natural hygroscopic characteristic of honey (dehydration of the bacteria by hyperosmolar properties) for *E.coli*, inhibition is related to physicochemical characteristics of honey and not necessarily to antimicrobial compounds.

### 2.2. Screening of Strains with Antimicrobial Activity

The 28 isolated strains of *S. aff. postica* and *A. mellifera* grew under anaerobic conditions in the absence of oxygen. With a concentration of less than 0.1% Oxygen. The strains were coded according to the bee species and the number of colonies isolated from the Petri dishes. Isolated strains of *S. aff. postica* were coded as SCA, and the isolated strains of *A. mellifera* were coded as API. The 28 isolated strains underwent an antimicrobial selection test, in which their ability to inhibit pathogenic bacteria *B. cereus* (Biomedh 11778, Itapoa, Belo Horizonte, *E. coli* (Laborclin 25922, Pinhais, Brazil), *S. aureus* (Laborclin 25923, Pinhais, Brazil) and *Salmonella* sp. (Biomedh 14028, Itapoa, Belo Horizonte) was tested. The tests were performed in triplicate and the interactions were analyzed according to the presence or absence of halos. Table 2 shows the results of the presence or absence of inhibition by the isolated strain, classified into low (<11 mm), medium (12–14 mm) and high (>15 mm) activity, respectively, according to the size growth inhibition halo in the antimicrobial test. From a qualitative analysis, halos with small diameters showed low antimicrobial activity, as well as halos with large diameters, showed high antimicrobial activity.

*Bacillus cereus* was inhibited by 14 isolates, highlighting the SCA13 strain, which showed greater inhibition than the positive control. *E. coli* was inhibited by eight strains, and *S. aureus* showed fewer strains with antimicrobial capacity, totaling seven isolates. *Salmonella* sp. was inhibited by ten honey isolates. The SCA-07, SCA-13, and SCA-15 strains showed the highest antimicrobial spectrum among those isolated from *S. aff. postica* honey.

From the results obtained, seven strains were selected from *S. aff. postica* and two from *A. mellifera*: SCA-07, SCA-11, SCA-12, SCA-13, SCA-15, SCA-16, SCA-17, API03, and API-06 according to the potential inhibitory activity against pathogenic bacteria (*B. cereus, E. coli*, *S. aureus*, and *Salmonella* sp.).

### 2.3. Morphological Identification and Molecular Identification

In the morphological identification step, a Gram stain was performed. The strains SCA-07, SCA-11, SCA-12, SCA-15, SCA-17, API-03, and API-06 were classified as Gram-variable because they contain Gram-positive bacilli at the top and Gram-negative cocci at the bottom. The SCA-13 strain was classified as a Gram-positive coccus. The SCA-16 strain was classified as a Gram-positive bacillus.

The SCA13 strain was selected according to the best results obtained by screening for strains with antimicrobial activity for molecular identification. Its total DNA was extracted using phenol/chloroform and characterized via a comparison of homology to the 16S gene. The DNA sequences obtained from the ACTGene Molecular Analysis were compared with existing sequences using the Basic Local Alignment Search Tool (BLAST) of the National Center of Biotechnology Information (NCBI), which resulted in a 98% match to the species *Enterococcus faecalis.* The result of the analysis of sequence similarities is in Appendix A.

### 2.4. Antagonist Activity

The bacteria isolated from honey selected in the inhibitory potential test underwent antagonist analysis and were tested with the spot-on-lawn assay to determine whether they had antagonistic activity against indicator bacteria, *B. cereus* and *L. monocytogenes* (Figure 2). Selected strains were tested, and antagonist activity was detected by the formation of a growth inhibition halo. In Figure 2, it is possible to observe inhibition halos against the indicator bacteria to select the antimicrobial-producing bacteria.

Strains SCA07, SCA16, SCA17, API03, and API06 were not significantly different from the control (*p* > 0.05) and strain SCA11 (*p* ≤ 0.05), that is, they showed lower inhibition than vancomycin, in relation to the indicator. The SCA12, SCA13, and SCA15 strains were statistically significant (*p* < 0.00001). Therefore, they had an inhibitory action similar to that of vancomycin, used as a positive control (Figure 3).

Later, lyophilized extracts isolated from SCA12, SCA13, and SCA15 were tested in wells by diffusion to determine whether the substances produced by the bacteria isolated from honey were produced extracellularly or intracellularly. SCA13 and SCA15 were able to inhibit both pathogenic bacteria, whereas the SCA12 strain was not able to inhibit *L. monocytogenes*. The SCA13 strain exhibited a higher inhibition halo, as shown in Table 3.

### 2.5. Extract Characterization

The exposed strain SCA13, which is facultative aerobic, showed an extract with higher inhibitory activity under anaerobic conditions against both pathogens (*B. cereus* and *L. monocytogenes*) than under aerobic conditions.

The inhibitory activity of SCA-13, a facultative anaerobic Gram-positive coccus (*E. faecalis*) showed higher inhibition at anaerobiosis treatment. Excess oxygen and oxidative stress can disrupt the growth of most organisms, but the underlying mechanisms of damage have proved difficult to unravel. For instance, O_2_^−^ and H_2_O_2_ can oxidize the exposed Fe–S clusters of a family of dehydratases. This event destabilizes the clusters, and their consequent disintegration eliminates enzyme activity. SCA-13 had to be able to overcome a number of barriers including anaerobiosis, pH shifts, and high osmolarity. The isolated bacteria from honey also had to develop successful strategies to be able to compete with other organisms for substrate found in these environments. In addition, SCA-13 must also assimilate nutrient substrates as well as survive under the harsh conditions presented to it in honey and in the gastrointestinal tract of bees.

With the metabolites of the inoculum growth of the SCA13 strain having been more favorable in anaerobiosis, the lyophilized anaerobic extract underwent three types of treatments (thermostability, sensitivity to neutral pH, and sensitivity to proteolytic enzymes).

In relation to temperature sensitivity, the extract of strain SCA13 was autoclaved at 121 °C for 15 min. The extracts obtained after autoclaving did not show statistical differences from vancomycin. The supernatant extract of the SCA13 strain was incubated for 72 h to inhibit the growth of the indicator bacteria from the thermostability treatment at 121 °C. From Table 4, it can be inferred that the proteins were not denatured during heat treatment, as they did not present statistical significance (*p* > 0.05) for the two indicator bacteria; that is, the extract of the SCA13 strain was not altered in the thermostability test at 121 °C, which shows the resistance of antimicrobial peptides to high temperatures.

Inhibition halos formed in the neutral pH (pH 7.0) characterization of SCA13 strain extracts. The extract of the SCA13 strain did not show statistical significance (*p* > 0.05) against either pathogen, so it can be stated that the neutralization of the extract was not able to reduce the antimicrobial activity of the peptides.

As for sensitivity to the proteolytic enzyme trypsin, the extract did not show a significant difference (*p* > 0.05) between *B. cereus* and *L. monocytogenes*, showing that it was not inactivated after a possible inactivation with the enzyme trypsin in relation to the control without any type of treatment. The inactivation of antimicrobial peptides in the presence of trypsin and other proteolytic enzymes would be an indication that the substance is a protein in nature.

The lyophilized extract of the SCA13 strain that showed inhibitory activity in the well diffusion test was tested with the application of treatments to determine its characteristics and spectrum of action under different conditions. Table 4 presents the results.

## 3. Discussion

Every day, new antimicrobials are sought to evade antimicrobial resistance generated by the abusive and inappropriate use of antibiotics. Honey is a natural, healthy product with medicinal properties and antimicrobial activity, closely related to its physicochemical characteristics and, more recently, to the gut microbiota of honeybees. In the present study, we found that the higher the concentration of honey, the greater the inhibition of pathogenic bacteria. It is important to emphasize that the initial results were not able to verify the hypothesis that the bacteria present in the honey were responsible for the inhibition owing to the chemical and nutritional composition of the honey, which contains other compounds capable of inhibiting it. The water content of honey is undoubtedly one of the most important characteristics, as it influences its viscosity, specific weight, maturity, crystallization, flavor, conservation, and palatability [22].

Many species of microorganism are found in the intestines of bees; of these 1% yeast (*W. anomalus*), 29% are gram positive bacteria, including *Bacillus*, *Bacterium*, *Streptococcus* and *Clostridium* species, and 70% are gram-negative or gram-positive bacteria, including *Achromobacter, Citrobacter*, *Enterobacter*, *Erwinia*, *E. coli*, *Flavobacterium*, *Klebsiella*, *Proteus* and *Pseudomonas.* [23]. These insects need to defend themselves against pathogens, but they also need to protect their food sources from attacks by microorganisms. According to Menegatti [24], one of the defense strategies acquired during the evolution of insects is symbiotic association with bacteria capable of biosynthesizing natural products (antibiotics and antifungals) against pathogens. Therefore, most of the bacteria found in both the honey of *A. mellifera* and stingless bees have antimicrobial capacity as a function of the bees’ defense against pathogens found in their natural habitat.

The SCA13 strain resulted from the molecular identification of *Enterococcus faecalis,* a species of bacteria that is abundant in food sources of animal origin such as cheese, chicken, and cassava [25]. In addition, it survives under adverse conditions such as extreme pH, temperature, and salinity [26]. They can produce substances with antagonistic potential, such as organic acids (lactic acid), hydrogen peroxide, lytic bacteriophages, proteolytic enzymes, and antimicrobial substances of a protein nature, called bacteriocins [27,28]. Studies have been developed on the antibacterial activity of honey on antibiotic-resistant pathogens, pathogenic bacteria involved in some diseases, pathogenic bacterial food and bacteria responsible for the deterioration of food (spoilage bacteria), and their different levels of sensitivity to honey. Bacteria such as *Staphylococcus aureus*, *Staphylococcus epidermidis*, *Bacillus stearothermophilus* are extremely sensitive, while *Staphylococcus uberis*, *Escherichia coli*, *Klebsiella pneumoniae*, *Bacillus cereus*, *Alcaligenes faecalis*, *Lactobacillus acidophilus*, *Helicobacter pylori*, *Bacillus subtilis* are moderately sensitive and the growth of *Micrococcus luteus*, *Enterococcus faecalis* and *Pseudomonas aeruginosa* appear to be unaffected by honey [29].

Using the spot-on-lawn assay, Nardi [30] observed a better comparison of antimicrobial activity, according to Çadick and Çitak [31], these metabolites can be produced during the test period, which can increase the diffusion capacity of the substances produced by the bacteria and the concentration of these substances in the culture medium with three strains that showed inhibitory activity against the two indicator bacteria.

These extracts were tested by diffusion in wells to determine whether the substances produced by the bacteria isolated from honey were produced extracellularly or intracellularly. Studies carried out on *Solanum trilobatum* leaves resulted in a protein with antibacterial potential against *V. cholerae* and *S. aureus* evaluated by well diffusion method and Minimum Inhibitory Concentration (MIC) by microdilution assay. In addition, the protein was identified by peptide mass fingerprinting, and morphology by the three-dimensional structure [32]. In a study by Salles [9], many strains isolated from açai showed growth inhibition halos using the spot-on-lawn technique against *B. cereus* and *L. monocytogenes*. These results agree with those of the honey strains, in which the extract of the SCA13 strain showed the highest inhibitory rates in the well-diffusion test through growth inhibition halos among others, as shown in Figure 2. According to Dimitrieva-Moats and Unlu [33], lyophilized extract promotes more stable activity. Thus, inhibition was observed due to an increase in the concentration of the extract produced. Using the well-diffusion technique, it was possible to verify that the metabolites produced by the bacteria isolated from honey were produced extracellularly. Table 3 shows that the selected producing bacteria were able to inhibit the two indicator bacteria, except for the SCA12 strain, which was not able to inhibit *L. monocytogenes*.

The characterization tests of the bacterial extract were carried out because some physicochemical factors (such as pH and acidity) are considered important antimicrobial factors, providing greater stability to the product regarding the development of microorganisms. According to studies by Nogueira-Neto [34], the optimal pH for the growth range of several pathogenic microorganisms in animals is between 7.2 to 7.4. Melipona honey has high acidity, which is an indicator of the tested antimicrobial properties. Many authors have claimed that some bacterial strains increase the production of metabolites when they are subjected to a certain level of stress. This explains the increased production of peptides due to oxidative stress or in a microaerophilic environment. According to Ochner [35], several proteins are produced by bacteria in the defense against oxidative stress, many of which are related to the response to this stress have already been identified, although there is a lack of studies that explain their mechanisms of action in a more complex way.

Regarding the concept of sensitivity to the proteolytic enzyme trypsin, some studies, such as that conducted by Salles [9], showed resistance to the trypsin enzyme in two types of treatment times: 2 h and 18 h. In contrast to the work by Bromberg [36], the bacteriocin produced by CTC 484 culture showed saline sensitivity to trypsin. These results are in concordance with the extract of the SCA13 strain subjected to trypsin treatment, which showed growth inhibition. Govan and Harris [37] stated that extracts that were reactive in the presence of trypsin might be new types of peptides that have not yet been studied.

Breeding native stingless bees is a prominent activity in the state of Pará, with many species indicating potential for management and breeding in the Amazon biome [38]. The breeding of “straw bees” (*S. aff. postica*) has recently gained prominence because of its potential use in directed pollination of the açai palm tree (*Euterpe oleracea*) crop, and productivity could be increased by up to 2.5 times with the pollination service [39]. The results of this study allow us to infer the importance of honey in the creation of this bee species, and this product may become a future tool for bioactive compounds with potential uses in the pharmaceutical industry.

## 4. Materials and Methods

### 4.1. Collection, Asepsis, and Isolation of Strains

Honey samples from *S. aff. postica* and *A. mellifera* were collected from the municipality of Santa Maria do Pará in 2018. The honey samples from both species used in this study, *A. mellifera* and *S. aff. postica*, showed floral predominance in açai (*Euterpe oleracea*) cultures. It was found that the honey was monofloral for açai in *S. aff. postica* honey. Four samples (A, B, C, and D) were collected from *S. aff. postica* and five samples (A, B, C, D, and E) were collected from *A. mellifera*. Honey was collected using a syringe and stored in 50 mL falcon tubes. The samples were stored at the Center for Valorization of Bioactive Compounds of the Amazon (CVACBA). Nine samples were used: four (A, B, C, and D) of honey from *S. aff. postica* and five samples (A, B, C, D, and E) of honey from *A. mellifera.*

From these samples, 30 g of honey was used in three concentrations: the first pure reference to a concentration of 100%, the second was a concentration diluted in 50% saline solution (FUJIFILM Irvine Scientific 0.85%, Santa Ana, CA, USA), and the third corresponded to a concentration diluted in 25% of honey in saline solution (FUJIFILM Irvine Scientific 0.85%, Santa Ana, USA) to evaluate whether the concentration of honey is a parameter that influences the number of colonies obtained. Subsequently, the techniques of spread plate and pour plate of the concentrations obtained in Petri dishes containing culture medium agar of Man Rogosa Sharpe (MRS agar) were performed to verify whether the bacteria present in the honey had aerobic or anaerobic characteristics. The procedure was carried out in 12 plates, divided into spread plates (100%, 50%, and 25%) for aerobiosis, spread plates (100%, 50%, and 25%) for anaerobiosis, pour plates (100%, 50%, and 25%) for aerobic, and pour plates (100%, 50%, and 25%) for anaerobic conditions. The plates were then incubated at 37 °C for 72 h.

With the acquisition of four samples of honey from *S. aff. postica*, serial dilutions were performed in up to 10 units of honey in peptone saline solution (HIMEDIA, Model AG-7013/SP, West Chester, PA, USA) for each sample, totaling 20 plates. Soon after dilution, the honey solutions were placed in Petri dishes containing Man Rogosa and Sharpe (MRS agar) using the spread plate technique. The solutions were spread over the culture medium using a Drigalski spatula. The plates were then incubated at 37 °C for 72 h under anaerobic conditions.

### 4.2. Screening—Antimicrobial Test

To perform the antimicrobial test, different concentrations of honey were initially used (100%, 50%, and 25%) against the pathogenic bacteria, *Escherichia coli* (Laborclin 25922, Pinhais, Brazil). This technique was carried out as follows: plates containing Mueller-Hinton agar prepared in advance were removed from the refrigerator until they reached room temperature. Wells (4 mm in diameter) were made on an agar plate. Using a sterile swab, the bacterial inoculum with 0.5 turbidites on the MacFarland scale was evenly distributed over the agar surface and left to rest at 27 °C for approximately 3 min. Fifty microliters of the concentrations obtained from honey from *A. mellifera* and *S. aff. postica* were dispensed into each properly identified well. After this procedure, the plates were incubated in a bacteriological oven at 37 °C for 24 h in aerobiosis with opening jar and anaerobic environment using a sealable jar with the BD GasPak™ EZ Anaerobe Container System (Sparks, USA).

To screen for candidate bacteria, another antimicrobial test was performed according to the methodology adapted from Patel [40], which consisted of filtering the metabolites produced by the honey isolates. The procedure consisted of applying a bacterial inoculum with a turbidity of 0.5 on the MacFarland scale using the turbidimetric method to standardize the pathogenic bacteria used in the test (*S. aureus*, *E. coli*, *B. cereus*, and *Salmonella* sp.). The inoculum was distributed over the surfaces of Petri dishes containing Mueller-Hinton agar medium. Wells were made on the surface of the agar, 4 mm in diameter, and agar was removed from the wells with sterile forceps. The wells were properly identified, and 25 µL of bacterial filtrate from colonies isolated and cultivated from honey from *A. mellifera* and *S. aff. postica* was dispensed into the wells according to the strains isolated. The plates were then incubated in an oven at 37 °C for 24 h. The antibiotic was Streptomycin (10 µg) (Laborclin 640623, Pinhais, Brazil). All selected positive samples were analyzed in triplicate.

### 4.3. Identification of the Genera of Microorganisms

The identification was based on physicochemical tests following the methodology of ANVISA [41] for the detection and identification of bacteria of medical importance.

#### 4.3.1. Genomic DNA Extraction

Total DNA was extracted from the bacterial DNA extraction as described by Seldin and Dubnau [42] using phenol-chloroform. After extraction, the DNA sample was mixed with blue juice dye in a proportion of 1:2 of the total volume of the mixture and subjected to horizontal electrophoresis in 1 % (*w*/*v*) agarose gel. Ethidium bromide (0.1%) was added to 10× diluted Tris-acetate-EDTA (TAE) buffer for 30 min at a constant voltage of 90 V. The molecular weight marker used was the “1 Kb Plus DNA Ladder” (Promega^®^, Madison, USA). The gel was observed and photographed under UV light using a UV light transilluminator (model) to analyze the purity and quality of the genetic material.

#### 4.3.2. Molecular Identification

The bacterial strain was molecularly identified by amplifying the entire 16S rRNA gene fragment using universal primers 8F (5’-AGAGTTTGATCCTGGCTCAG-3’) [43] and 1492R (5’-GGTTACCTTGTTACGACTT-3’) [44]. PCR was performed in a final volume of 70μL containing 5X Green GoTaq^®^ Flexi Buffer (1X) from Promega^®^, 1 mM MgCl_2_, 0.2 mM dNTPs, 1.5 μM of each primer, 5 U/μL of GoTaq^®^ DNA polymerase, DNA at concentrations between 50 and 100 ng/μL, and sterile filtered water (all reagents from Sigma-Aldrich^®^). The cycle applied was: 35× (1 min at 95 °C; 1 min at 55 °C; 1 min at 72 °C); 1× (10 min at 72 °C); 4 °C. Sequencing of the DNA sample was performed through outsourced services of the company ACTGene Molecular Analysis using the Sanger method [45]. The obtained sequences were compared with those present in GenBank using the Basic Local Alignment Search Tool (BLAST) of the National Center of Biotechnology Information (NCBI) using the BLAST nucleotide tool (BLASTn) [46].

### 4.4. The Antagonist Activity

#### 4.4.1. Spot-on-Lawn Assay

The bacteria that showed antimicrobial activity and were consequently selected for the screening process underwent an antagonist test against other microorganisms called indicator cultures (*Bacillus cereus* and *Listeria monocytogenes*).

According to the method described by Nardi [30], bacterial samples were activated in test tubes containing MRS broth (Man, Rogosa, and Sharpe) and incubated at 37 °C for 24 h under aerobic conditions, shaking the tubes in the Shaker TOS20 orbital shaker at 200 rpm. After activation, 5 µL of each sample was deposited in the center of a petri dish containing MRS agar and incubated in an anaerobic chamber at 37 °C for 24 h. After this period, chloroform was added to the lids of the Petri dishes with sterilized cotton under UV light for 30 min to eliminate microorganisms. Immediately after UV exposure, the plates were closed under laminar flow. The indicator cultures underwent activation and were grown in brain-heart infusion (BHI) broth. Soon after, 10 µL of the indicator culture was added to tubes containing 3.5 mL of semi-solid agar. Then, the tubes with semi-solid agar containing the indicator bacteria were poured into Petri dishes, with 3.5 mL of semi-solid agar in 15 mL Falcon tubes added to the plates. Vancomycin-640630-Laborclin (30 µg) was used as the positive control. The plates were then incubated at 37 °C for 48 h under aerobic conditions. Inhibition zones were based on the Clinical and Laboratory Standards Institute—Performance Standards for Antimicrobial Susceptibility Testing method [47]. From the measurement of the size of the zone of inhibition with a halometer and the classification of strains into sensitive, moderately sensitive, intermediate, or resistant according to the diameter of the standard zone established for each antimicrobial drug. The analysis was performed in triplicate.

#### 4.4.2. Diffusion Test in Wells

The well diffusion test was carried out with the aim of determining whether the compounds produced by the bacteria isolated from honey present intra-or extracellular activity to facilitate the characterization process that was carried out later, as mentioned in the work performed by Salles [9]. Patel’s study [40] was used as a reference for the development of the analysis. Producing bacteria that showed inhibitory activity were activated in MRS broth and incubated at 37 °C for 24 h under aerobic conditions. The isolated bacteria were centrifuged at 5000 rpm for 5 min, and the supernatants were collected. The supernatant extracts of each producer bacterium were filtered through whatmann paper nº 2 with the help of a vacuum pump (Prismatec 132, Itu, Brazil) and then frozen at −20 °C for 24 h to be lyophilized (JJ cientifica, São Carlos, Brazil) for 72 h. After this process, the lyophilized content of each tube was resuspended in 500 µL of ultrapure water, and it was inoculated in wells made in BHI agar with 3.5 mL of semi-solid BHI medium and 40 µL of the indicator culture (*B. cereus* and *L. monocytogenes).* The plates were then incubated at 37 °C for 24 h under aerobic conditions. Extracts with activity were selected for characterization. The analysis was performed in triplicate.

### 4.5. Extract Characterization

The extracts produced by the bacteria that showed antimicrobial activity were characterized in their spectrum of action through four types of treatments [48].

#### 4.5.1. Oxygen Concentration

The extract of the bacterium that showed antimicrobial activity in the analysis of antagonist activity was tested to determine if the concentration of oxygen in the culture medium could interfere with the inhibition potential of the extract and subsequently optimize the production process of the lyophilized extract. For this, two different cultivations were carried out, one cultivation under aerobic conditions using agitator equipment to control the distribution of oxygen in the jars, and the method used to generate an anaerobic environment was the BD GasPak™ EZ Anaerobe Container System (Sparks, USA).

#### 4.5.2. Thermal Stability

Regarding thermal stability, the first step involved autoclaving the supernatant extracts at 121 °C for 15 min, after which the extracts were frozen and lyophilized. The next step consisted of resuspending the extract in 1000 µL of ultrapure water and then inoculating it into wells made in BHI agar culture medium with the addition of 3.5 mL of semi-solid BHI medium and 40 µL of the indicator bacteria (*B. cereus* and *L. monocytogenes*).

#### 4.5.3. Proteolytic Enzyme Sensitivity

Sensitivity to proteolytic enzymes was determined according to the method adapted [9]. The lyophilized extracts were resuspended in 1 mL of Tris-HCl buffer (50 mM, pH 7.6) plus calcium chloride (1 mM), with the proteolytic enzyme trypsin (Sequencing Grade Modified 9PIV511-PROMEGA modified) added to the medium at a ratio of 1:100 (12.5 ng/µL) and it was diluted in 100 µL of acetic acid buffer. The samples were then placed in a water bath at 37 °C overnight. Subsequently, it was inoculated into wells made in BHI agar with 3.5 mL of semi-solid BHI medium and 40 µL of the indicator (*B. cereus* and *L. monocytogenes*) culture. The plates were then incubated at 37 °C for 24 h under aerobic conditions [49].

#### 4.5.4. pH Effect

For pH characterization, the extracts were neutralized to pH 7.0, using sodium hydroxide (20%). The pH measurements were performed using pH indicator strips. After neutralization of the extracts, they were resuspended in 1000 µL of ultrapure water and later were inoculated in wells made in BHI agar added to 3.5 mL of semi-solid BHI medium with 40 µL of the indicator culture (*B. cereus* and *L. monocytogenes*).

### 4.6. Statistical Analysis

Statistical analyses were performed using the BioEstat^®^ software (version 5.0). Analysis of variance was performed using the Kruskal-Wallis test to verify whether the isolated strains had superior or inferior inhibitory action compared to the control (antibiotic). In all tests, a significant level of 0.05 (α = 0.05) was considered.

## 5. Conclusions

This study shows that bacteria isolated from the stingless bee honey, *Scaptotrigona aff. postica*, have the potential to produce antimicrobial substances. Of the twenty-eight bacteria isolated, three showed extracellular activity in the well-diffusion test after the lyophilization process, including SCA13 (which was identified as *Enterococcus faecalis*). Characterization tests are fundamental to defining the characteristics that differentiate antimicrobial peptides from others. The extract of the characterized bacteria showed a small decrease in activity in the presence of trypsin compared with the indicator bacteria. The results demonstrate the possibility of the substance contained in the supernatant extract being of a protein-like nature. The extracts showed activity after autoclaving at 121 °C, indicating that they may be class I and II bacteriocins. The neutralized extracts showed the same activity as that of the control extract, a factor that indicates better activity of the extract at acidic pH.

## Figures and Tables

**Figure 1 antibiotics-12-00223-f001:**
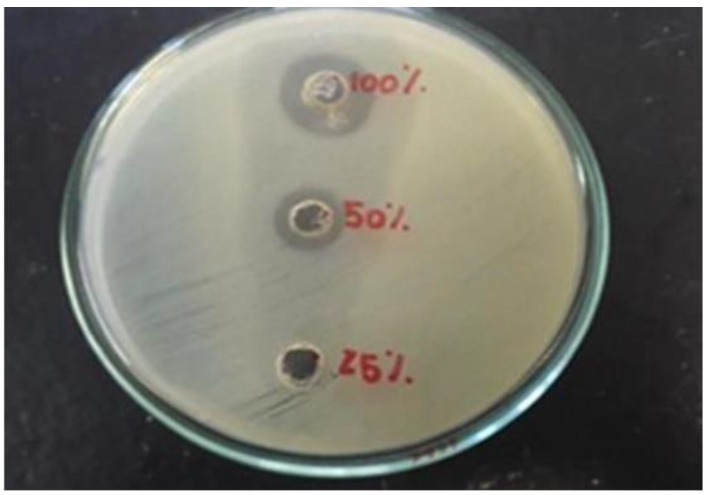
Antimicrobial test with different concentrations (25%, 50%, and 100%) of *S. aff. postica* honey against *E. coli*.

**Figure 2 antibiotics-12-00223-f002:**
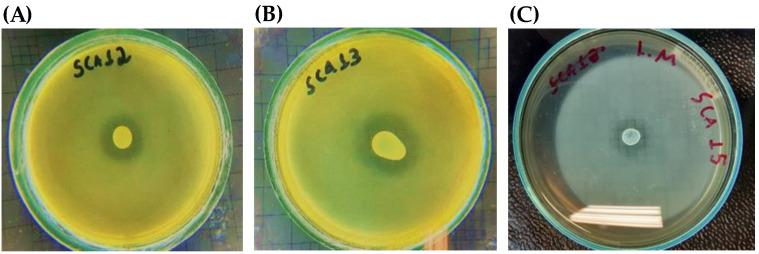
Spot-on-lawn assay results. (**A**) SCA12 bacterium showing growth inhibition halo against the indicator bacterium, *B. cereus*; (**B**) SCA13 bacterium showing growth inhibition halo against the indicator bacterium, *B. cereus*; (**C**) SCA15 bacterium showing a halo of inhibition against the indicator bacterium, *L. monocytogenes*.

**Figure 3 antibiotics-12-00223-f003:**
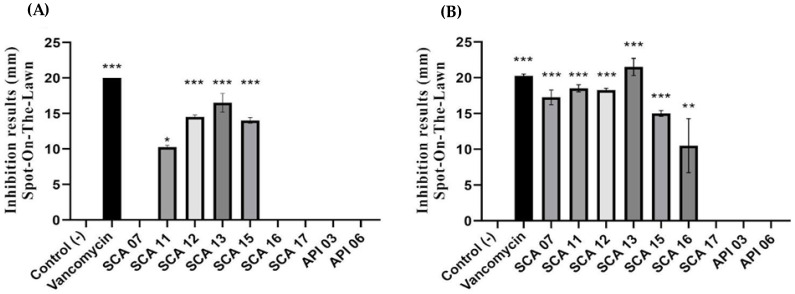
Antimicrobial activity of strains isolated from different kinds of honey against (**A**) *L. monocytogenes*, and (**B**) *B. cereus*. * *p* < 0.05; ** *p* < 0.01; *** *p* < 0.00001.

**Table 1 antibiotics-12-00223-t001:** Microbial growth of *A. mellifera* honey samples on MRS Agar for aerobic and anaerobic processes.

Presence of Oxygen	Honey Concentration	Spread Plate	Pour Plate
Aerobiosis	100%	−	−
Aerobiosis	50%	−	−
Aerobiosis	25%	−	−
Anaerobiosis	100%	−	+
Anaerobiosis	50%	+	+
Anaerobiosis	25%	+	−

**Table 2 antibiotics-12-00223-t002:** Antimicrobial selection test of strains isolated from the honey of *S. aff. postica* (SCA) and *A. mellifera* (API). (−) No activity, (+) Low activity, (++) Medium activity, (+++) High activity.

Isolated Strain	Inhibitory Activity			
	*B. cereus*	*E. coli*	*S. aureus*	*Salmonella* sp.
SCA01	−	−	−	−
SCA02	−	−	−	−
SCA03	−	−	−	−
SCA04	−	−	−	+
SCA05	−	−	−	+
SCA06	−	−	−	++
SCA07	++	+	++	++
SCA08	−	−	+	−
SCA09	−	−	−	−
SCA10	−	−	−	−
SCA11	++	++	−	+++
SCA12	+	++	−	++
SCA13	+++	++	+	+
SCA14	+	+	−	−
SCA15	+	++	+	++
SCA16	++	−	++	−
SCA17	++	+	−	++
SCA18	+	−	−	++
SCA19	+	−	−	−
SCA20	−	−	−	−
SCA21	−	−	−	−
SCA22	−	−	−	−
API01	−	−	−	−
API02	+	−	−	−
API03	++	+	−	−
API04	−	−	−	−
API05	+	−	+	−
API06	++	−	+	−
Streptomycin	++	++	+++	+++

**Table 3 antibiotics-12-00223-t003:** Well-Diffusion test results of lyophilized isolated extracts. (−) No activity, (+) Low activity (<11 mm), (++) Medium activity (12–14 mm) and (+++) High (>15 mm) activity.

Isolated Strain	Inhibitory Activity	
	*B. cereus*	*L. monocytogenes*
SCA12	++	−
SCA13	+++	++
SCA15	+	+

**Table 4 antibiotics-12-00223-t004:** Treatment results. Mean and standard deviation of growth inhibition halos (with 10 mm). * SCA13 without Treatment.

Strain SCA13 (Treatment)	Inhibitory Activity	
	*B. cereus*	*L. monocytogenes*
Control wT *	10.25 ± 0.4	10.25 ± 0.5
Autoclaved	11.50 ± 0.5	11.00 ± 0.0
Neutral pH	10.25 ± 0.4	08.50 ± 0.5
Trypsin	08.00 ± 0.0	08.00 ± 0.0

## Data Availability

All data generated or analyzed during this study are included in the published article.

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
