# Peer review of "Evaluation of the Antimicrobial Capacity of Bacteria Isolated from Stingless Bee (Scaptotrigona aff. postica) Honey Cultivated in Açai (Euterpe oleracea) Monoculture"

_antibiotics, 2023, doi:10.3390/antibiotics12020223_

Round 1

Reviewer 1 Report

The article titled “Evaluation of the antimicrobial capacity of bacteria isolated
from stingless bee (Scaptotrigona aff. postica) honey cultivated
in (Euterpe oleracea) monoculture” attempted to find some antimicrobial metabolites from the two bees, though the topic is interested and the amount of results is not qualitative. Here I am providing my comments with major revision.

1.      In line 97 table description should write below the table and should align properly

2.      All the tables should be changed as mentioned above

3.      In Fig.2, it is so difficult and doubtable to differentiate between cocci and bacilli on staining background.

4.      Line: 167 need the sequence data result since the gram stain result not enough to make conclusion.

5.      Author telling the metabolites or peptides were evidenced the antimicrobial activity from Scaptotrigona aff. Postica, the isolated extracts but author did not provide sufficient information like peptide information or mass spectrometry or any other supporting materials to strengthen the data without the detail information it is very difficult to make a decision.

6.      Author efforts and the conducted research were really more important and interesting but the results what here provided also not meeting the scientific standards, it would be more useful if author improve the manuscript.

Author Response

RE:  [Antibiotics] Manuscript ID: antibiotics-2050887 - Major Revisions (Please Revise on the Layout Version Attached

I refer to your letter of November 24th, 2022, with your comments and those of reviewers on our manuscript. The authors would like to thank the reviewers for the precise and insightful comments that markedly improved the quality of the manuscript. We have addressed all the reviewer’s comments on a point-by-point basis. The answer to each addressed Reviewer’s comment was followed in red, and the included text was highlighted in yellow on the manuscript

We hope that the revised manuscript will be acceptable in Antibiotics, and we thank you in advance for your consideration

Sincerely yours,

Reviewer 2 Report

Even though the topic of isolation of antimicrobial compounds is interesting, I have to reject this study. The discussion and materials and methods are poor. Also, the results could be better. Only one isolate was properly identified. And the suggested strain belongs to a species which also has issues and safety assessment will be tough.

Even though the English is not bad, the text is hard to read since the authors jump from one topic to another topic in one line.

Additionally, I would not talk about pharmaceutical application at this stage of the studies.

See my comments below

Abstract:

Line 13: honey bacteria don’t exist. Use bacteria isolated from honey instead

Introduction

Line 73-74: Bacteria produce acids in primary metabolism, not secondary. Or how do you mean secondary manner here?

Line 76: I assume composts is the wrong word here.

Line 80-82: why are mesophilic bacteria probiotic candidates? If so, thermophilic would be better candidates. However, it is not valid to talk about probiotic candidates in this context anyway. IIt needs way more properties to fulfill this definition.

Line 86: I would narrow pathogenic bacteria already here to which species/group you selected.

Results:

Line 91: How many samples?

Line 94-96: Why was no growth detected in aerobic plates? Why did you use only MRS? What do the percentages mean? It is described in the table but not the text.

Line 100-107 and Table 2: Not needed. One or 2 sentences indicating low bacterila content is enough.

Line 108-116: Which was the indicator organism? Mention it also in the text.

Line 110-115: May due to osmotic stress of sugar content? It is mentioned in discussion but I would also make a point here.

Line 121-122: Presence of oxygen is by definition not anaerobic. I assume you mean microaerophilic?

Line 128-129: You indicate that Table 3 only evaluates presence or absence of inhibition. The table itself however classifies. How did you define the levels?

Line 137: Which other pathogens? The used ones in this study? No need to compare then.

Line 140-142: Reason to select them?

Line 143: How do you define the gender of a bacterium?

Line 145-147: Did you see spores in the microscope? Clostridia are usually very anaerobic, some are even killed in the presence of oxygen. Please elaborate how a bacterium can be a gram-positive bacilli at the top and a gram-negative cocci at the bottom.

Line 143-169: Why did you not use better identification methods? MALDI-TOF, 16S? Even API would be a good indication. Only looking at image 2C I doubted that it is a Staph. Spp. 16S confirmed it. Please use proper identification of bacteria if you want to publish a paper like this. At least the 4 strains in Figure 2 should have been properly characterized.

Line 173: Revealing is not the right word. Indicator organism or bacteria. Which strains did you use? Reference strains? Isolates from clinical cases?

Line 210-213: How is a facultative aerobic bacteria stressed under anaerobic conditions? How can you argument like this? This conclusion is wrong.

Line 228-230: Why should a lower pH reduce inhibition values in Listeria? Isn’t it the case that foods with lower pH are safer? Even with Listeria? What was the pH after neuztralization?

Line 232-234: What has neutralization too do with trypsin? You make a huge jump here.

Table 7: How do you explain the increased inhibition after autoclaving?

Discussion

Line 254-257: 29% gram-positive and 70 % gram-positive and gram-negative? How does this work?

Line 265-270: What you don’t mention is, that it is a facial indicator in dairy, able to have virulence factors, intrinsically produces biogenic amines and transfers antibiotic resistance genes very easily.

Line 272-281: Not a discussion but description. Make it shorter.

Line 287: define satisfactory

Line 288-290: This is not true. Longer chain organic acids won’t evaporate. Also, lactic acid can also be measured after freeze drying

Line 300-305: No idea how this fits here and what the authors want to say.

Line 307-308: Again, in an anaerobic environment, there is no oxygen. Anaerobiosis is the definition of 0 % oxygen.

Material and Methods

Provider of chemicals?

Line 339-340: sterile saline? Which concentration? With peptone? How was it mixed?

Line 341-342: How much honey was used to dilute? Same saline above?

Line 343-344: It’s de Man Rogosa Sharpe

Line 350-355: Not a repetition of above? Drigalski loop, not spatula?

Line 356-363: There is nothing from this in the results/discussion section

Line 366-367: Which strains? Reference strains? Where to get them?

Line 373-374: Aerobic or anaerobic? How was the atmosphere maintained?

Line 419: Aerobic conditions? Did you shake the flasks?

Line 437-439: How were the conditions for freeze drying?

Line 446: How did you measure oxygen concentration?

Line 453-454: In your methods, I don’t see any difference in incubation conditions (aerobic/anaerobic) in your assays. You can’t say which oxygen concentration you had in your assays.

Line 462-469: Why didn’t you inactivate the Trypsin?

Author Response

(The authors gave the same response as above.)

Reviewer 3 Report

Congratulations on the work. The authors describe the antimicrobial activity of honey bacteria. It is a very interesting topic, but there are some aspects that could be improved.

Why do the authors argue that the antimicrobial activity is due to compounds produced by SCA and API bacteria and not due to the physicochemical characteristics of honey or antagonism between bacteria? Improve the explanation.

In relation above bacterial species used in antimicrobial assays, do the authors use type or clinical strains? I would confirm the antimicrobial activity with both.

It would be very interesting if the authors give more information about the compounds produced by the SCA13 strain… Is it possible to reach the final antimicrobial molecule and identify it? For example, by chromatography…

Improve the photography in figure 1 b. The strain is not evenly seeded and the agar looks broken.

Author Response

(The authors gave the same response as above.)

Round 2

Reviewer 1 Report

The article discussed about the isolated bacterias  anti microbial activity from stingles bee. The object of the study is interest to readers unfortunately the author could not show any  detailed evidences against the bacterias reasonable for the function (or) the provided preliminary results are not enough to meet the standard the journal. 

Author Response

Response to Reviewer 1 Comments

The article discussed about the isolated bacterias  antimicrobial activity from stingles bee. The object of the study is interest to readers unfortunately the author could not show any detailed evidences against the bacterias reasonable for the function (or) the provided preliminary results are not enough to meet the standard the journal. 

We thank the reviewer for the comments, and it is in our plans for the next studies to isolate and purify the antimicrobial molecule to properly identify it. To provide more information about bacteria with antimicrobial activity.

The present study comes from the cultivation of stingless bees in the Amazon,  a sustainable initiative to aggregate value in a product coming from the Amazon Forest. Studies that enlightened new applications of traditional honey cultivation could bring a new perspective to reduce deforestation and stimulate a green economy.

We thank the reviewer for your time and consideration.

Reviewer 2 Report

Unfortunately, the authors did not improve the article in a way that I can accept it. Therefore, I still reject it. See some additional remarks below.

Results

Line 102-103: I don’t understand the sentence in this context.

Line 119-121: This indicates that at least for e. coli, the inhibition is related to honey and not antimicrobial compounds.

Line 135: I don’t think my remark of the first review was understood. My question was, how you make the anaerobic atmosphere? Microaerophilic means, you have >5 % of oxygen, anaerobic <0.1 %. Absence of oxygen is not microaerophilic.

Line 144-145: Which diameter was low, medium, or high?

Table 3: Why is the range of inhibition halo that high for SCA16?

Line 195: Control is vancomycin? Make this clearer here.

Figure3: Decide to use Figure 3 or Table 3

Table 4: Again, which mm corresponds to low, medium, or high

Table5: Don’t talk about oxygen concentration if you didn’t measure it

Line 242-244: I don’t understand this fully.

Line 248-252: Reformulate the text so it’s clearer that the antimicrobial component was not sensitive to Trypsin.

Table 6: What is the diameter of the well you add the extract to?

Discussion

Line 272: 1% of all yeast species worldwide?

Line 273-275: no correction of the percentage gram-+ and gram - -. Are the 29 % included in the other 70%?

Line 310-312: I don’t understand this sentence. Does lyophilization produce extracts?

Material and methods

Line 387-388: in anaerobic jars, were there anaerobic bags placed? Else it’s not anaerobic

Line 433: Which shaker? Which speed?

Line 455: Which vacuum? Which T? Which time?

Line 486: How do you elute in buffer? Do you mean dilute? How did you inactivate the enzyme?

References:

1.)    Why not link to the journal? Are the authors allowed to publish it on researchgate?

Author Response

I refer to your letter of December 14th, 2022, with your comments and those of reviewers on our manuscript. The authors would like to thank the reviewers for the precise and insightful comments that markedly improved the quality of the manuscript. We have addressed all the reviewer’s comments on a point-by-point basis. The answer to each addressed Reviewer’s comment was followed in red, and the included text was highlighted in yellow on the manuscript

We hope that the revised manuscript will be acceptable in Antibiotics, and we thank you in advance for your consideration

Reviewer 3 Report

The authors have satisfactorily answered all my questions and improved the manuscript. 

Author Response

(The authors gave the same response as above.)

Round 3

Reviewer 1 Report

This manuscript is more interest to the readers and the effort would be more appreciable to identify and searching for antibacterial peptides, author can follow and cite the article (https://www.mdpi.com/1420-3049 /27/4/1167#metrics).

Since author improved this article now recommending the manuscript for acceptance.

Author Response

[Antibiotics] Manuscript ID: antibiotics-2050887 - Minor Revisions (Due Date: 30 December)

I refer to your letter of December 28th, 2022, with your comments and those of reviewers on our manuscript. The authors would like to thank the reviewers for the precise and insightful comments that markedly improved the quality of the manuscript. We have addressed all the reviewer’s comments on a point-by-point basis. The answer to each addressed Reviewer’s comment was followed in red, and the included text was highlighted in yellow on the manuscript

We hope that the revised manuscript will be acceptable in Antibiotics, and we thank you in advance for your consideration
